# Carbon Nanotube Fibers Decorated with MnO_2_ for Wire-Shaped Supercapacitor

**DOI:** 10.3390/molecules26113479

**Published:** 2021-06-07

**Authors:** Luman Zhang, Xuan Zhang, Jian Wang, David Seveno, Jan Fransaer, Jean-Pierre Locquet, Jin Won Seo

**Affiliations:** 1Department of Materials Engineering, KU Leuven, Kasteelpark Arenberg 44–bus 2450, B-3001 Leuven, Belgium; lumanzhang1990@gmail.com (L.Z.); david.seveno@kuleuven.be (D.S.); jan.fransaer@kuleuven.be (J.F.); 2Industrial Research Institute of Nonwovens & Technical Textile, College of Textiles & Clothing, Qingdao 266071, China; wj8958@163.com; 3Department of Physics and Astronomy, KU Leuven, Celestijnenlaan 200D, B-3001 Leuven, Belgium; jeanpierre.locquet@kuleuven.be

**Keywords:** carbon nanotube fiber, CNT–MnO_2_ hybrid composite, wire-shaped supercapacitor, flexible electrode

## Abstract

Fibers made from CNTs (CNT fibers) have the potential to form high-strength, lightweight materials with superior electrical conductivity. CNT fibers have attracted great attention in relation to various applications, in particular as conductive electrodes in energy applications, such as capacitors, lithium-ion batteries, and solar cells. Among these, wire-shaped supercapacitors demonstrate various advantages for use in lightweight and wearable electronics. However, making electrodes with uniform structures and desirable electrochemical performances still remains a challenge. In this study, dry-spun CNT fibers from CNT carpets were homogeneously loaded with MnO_2_ nanoflakes through the treatment of KMnO_4_. These functionalized fibers were systematically characterized in terms of their morphology, surface and mechanical properties, and electrochemical performance. The resulting MnO_2_–CNT fiber electrode showed high specific capacitance (231.3 F/g) in a Na_2_SO_4_ electrolyte, 23 times higher than the specific capacitance of the bare CNT fibers. The symmetric wire-shaped supercapacitor composed of CNT–MnO_2_ fiber electrodes and a PVA/H_3_PO_4_ electrolyte possesses an energy density of 86 nWh/cm and good cycling performance. Combined with its light weight and high flexibility, this CNT-based wire-shaped supercapacitor shows promise for applications in flexible and wearable energy storage devices.

## 1. Introduction

In recent years, the ever-increasing demand for smaller and faster electronic components has stimulated a lot of research in the development of suitable energy storage devices [1,2]. Energy storage is one of the key requirements for flexible devices in particular [3]. Supercapacitors are considered to be promising charge storage devices owing to their fast charge/discharge rate and long cycle life [4,5]. Wire-shaped capacitors have shown various advantages in the development of lightweight, reconfigurable, and wearable electronics. Among the potential materials, carbon nanotube (CNT) fibers are promising candidates for wire-shaped electrodes owing to their remarkable mechanical properties, low weight, high conductivity, and excellent chemical stability [6,7,8]. In CNT fibers, the one-dimensional structure of each individual CNT, with its high electronic conductivity, promotes rapid charge separation and transport [9]. The outstanding potential of CNT fibers for various fiber-based devices has been demonstrated, including for e.g., supercapacitors [10], solar cells [11], actuators [12], and biosensors [13]. CNT fiber electrodes have also been applied for biocompatible implantable supercapacitors [14], as well as for neurochemical monitoring [15]. However, the electrical double-layer capacitance of CNT fiber alone is rather limited (~10–20 F/g) [16] and is insufficient for commercial devices. In order to circumvent this bottleneck, doping [17] or integration of pseudocapacitance materials (such as RuO_2_ and MnO_2_) into CNT fibers is beneficial and leads to a substantial improvement of the electrochemical performance of CNT fiber-based electrodes [18]. Among the different pseudocapacitance materials, MnO_2_ is particularly attractive for large-scale production due to its low cost, abundance, and high theoretical capacitance (1370 F/g). Nevertheless, MnO_2_ shows drawbacks, such as low electronic conductivity, poor rate performance, and limited cycling ability [19]. Therefore, the combination of MnO_2_ and CNT fiber represents a promising alternative leading to enhanced supercapacitor performance.

Different methods have been developed to fabricate CNT–MnO_2_ hybrid composites [20]. One of the most widely reported methods is electrochemical deposition: Ren et al. [21] reported a CNT–MnO_2_ composite fiber (4.1 wt.% MnO_2_) with a capacitance of 0.019 mF/cm. A CNT–MnO_2_ yarn supercapacitor with a capacitance of 25.4 F/cm^3^ at 10 mV/s has also been reported by Choi et al. [22]. Nevertheless, due to the hydrophobicity of CNT fibers, it is difficult for an aqueous solution to enter the pores inside CNT fibers. As a result, the deposition of MnO_2_ is mostly restricted to the surface of CNT fibers, which leads to a limited loading of MnO_2_ and hence to poor pseudocapacitance. In addition, the resulting deposit also critically depends on the condition of electrochemical deposition [23]. In order to overcome these difficulties, in situ routes have been explored. For example, Xu et al. [24] employed the reaction between ethanol and KMnO_4_ solution to deposit MnO_2_ nanoparticles on CNT fibers, and Cui et al. [25] used Mn(CH_3_COO)_2_·4H_2_O/ethanol solution to decorate CNT arrays with Mn_3_O_4_ nanoparticles. Nonetheless, these CNT–manganese oxide composites still lack homogeneous distribution of decorating nanoparticles and suffer from low capacitance. So far, detailed studies of CNT wire electrodes after deposition of manganese oxide are still rare, especially those that report their surface and mechanical properties. Such studies could move the development of CNT–manganese oxide composite electrodes a significant step forward to wearable devices.

In this study, we successfully fabricated flexible CNT–MnO_2_ fibers by spontaneous deposition of MnO_2_ on CNT fibers using a direct redox reaction between CNTs and KMnO_4_ aqueous solution with various concentrations of KMnO_4_ and reaction times, which yielded different but homogeneous MnO_2_ loadings (mass ratio of MnO_2_/CNT fiber). The wettability measurement showed that the fabricated CNT–MnO_2_ fiber was hydrophilic, which is crucial for the electrochemical performance of supercapacitors [26]. Surface energy components of CNT–MnO_2_ were determined by using the acid–base model [27] and physical adhesion between CNT–MnO_2_ and two common polymer electrolyte matrices (PVA and PVDF) was predicted. This prediction provided the criteria for selecting PVA as the more appropriate polymer electrolyte matrix for the fabrication of the supercapacitor. This systematic study also demonstrates that various aspects of the fabrication process that have so far not been sufficiently considered in the literature have a significant impact on the surface and mechanical properties of CNT–MnO_2_ fibers and highlight that particular attention is required when preparing the fibers and supercapacitors.

A wire-shaped supercapacitor was fabricated by assembling two aligned CNT–MnO_2_ fibers and showed a specific capacitance of 17.1 F/g (or 621.8 µF/cm) at 1 A/g coupled with good cycling stability (100% capacitance retention after 1200 cycles). Together with a tensile strength comparable with that of commonly used fibers (e.g., cotton [28]), these results demonstrate the promising potential of wire-shaped supercapacitors composed of CNT–MnO_2_ fibers for use as energy storage devices integrated into an electronic circuit for future wearable and deformable electronics.

## 2. Results and Discussion

For the fabrication of the wire-shaped supercapacitors, CNT fibers were produced through dry-spinning from a CNT carpet grown on Si. Details about CNT growth and the dry-spinning procedure of CNT fibers can be found in our previous work [29]. A schematic overview of the fabrication procedure of the CNT fiber-based supercapacitor is given in Figure 1.

Figure 2 shows the morphology and chemical compositions of the fibers investigated by scanning electron microscopy (SEM) and energy dispersive x-ray spectroscopy (EDS). The raw CNT fiber consisted of CNTs clustered to strands with an average diameter of 50 nm and with large gaps between individual strands, due to the fact that the fiber was not densified. Their surfaces were clean and smooth. After the reaction with KMnO_4_, the surface of the CNT strands appeared serrated, with decorated nanoflakes (Figure 2b–e). With increasing KMnO_4_ concentration (Figure 2b,e) or reaction time (Figure 2c–e), the amount of MnO_2_ nanoflakes deposited on the fiber surface increased gradually. The weight percentage of MnO_2_ in the CNT–MnO_2_ composite reached 85.7% for the CNT–MnO_2_ fiber (10, 12). The MnO_2_ nanoflakes were homogeneously distributed along the CNT strands and were connected to each other, which significantly increased the specific surface area of the fiber and was beneficial for improving the electrochemical performance [30].

Transmission electron microscopy (TEM) and scanning TEM (STEM) images confirmed the presence of MnO_2_ nanoparticles while the original shape of the raw CNT fiber was retained. The fiber was homogeneously covered with MnO_2_ nanoflakes, which appeared as bright compared to the dark-appearing CNTs in the annular dark field (ADF) STEM image in Figure 3a. These results were in agreement with the SEM results shown in Figure 2d. The size of the MnO_2_ nanoflakes can be estimated to be approximately 2–3 nm, as revealed by the high-resolution TEM image in Figure 3b. These small MnO_2_ flakes are particularly favorable as they can shorten the ion-diffusion length during the electrochemical reaction (compared to bulk MnO_2_), facilitating full utilization of the capacitance of MnO_2_.

It can be noted that the volume of the CNT fibers was reduced after reaction with KMnO_4_ solution and annealing at 200 °C. This shrinkage was caused by the elastocapillary aggregation of CNT strands and suggests the effective penetration of KMnO_4_ solution between the CNT strands and the effective wetting of the CNT surfaces [31], which also explains the homogeneous distribution of MnO_2_ flakes observed by SEM and STEM. As can be seen in Figure 2b–e, some of the CNT strands clustered together into larger ones, but large gaps between strands remained after the reaction, which are beneficial for electrolyte ion transportation [25]. In addition, with the highest MnO_2_ loading (Figure 2e), significant charging effects were noticed during SEM observations, indicating a reduced electronic conductivity. This was mainly a result of the low conductivity of the manganese oxide species densely decorating the CNT surfaces. The EDS results shown in Figure 2f reflect the chemical composition of the CNT–MnO_2_ fiber (10, 8), with the clear appearance of C, O, Mn, and K peaks confirming the presence of manganese oxides. The peak corresponding to K could originate from KMnO_4_ forming residual K_2_CO_3_ on the fibers [28].

Another observation is that the fiber became more fragile with increasing KMnO_4_ concentration and reaction time. This can be attributed to the deteriorated crystalline structure of CNTs after being exposed to KMnO_4_ [32]. For instance, CNT–MnO_2_ fiber (10, 12), with a 12 h reaction time, became very fragile and difficult to handle. Therefore, the reaction time was limited to a maximum of 12 h. Stress–strain curves of the raw CNT fibers and CNT–MnO_2_ fiber (10, 8) in Figure 4 confirmed this trend. The strain-to-failure and strength were strongly affected by the KMnO_4_ treatment. The 8 h treatment led to about 50% degradation in strength, indicating that the graphitic structure of CNTs is strongly affected by KMnO_4_.

Figure 5b–d shows Raman spectra taken from CNT–MnO_2_ fibers produced with 10 mM KMnO_4_ with 4 h, 8 h, and 12 h reaction times. The raw CNT fiber was used as a reference (Figure 5a). All spectra show a clear D-band at ~1350 cm^−1^ and a G-band at ~1585 cm^−1^, which correspond to disordered and graphitized carbon bonding in CNTs, respectively. The ratio of intensities of the G- and D-bands (*I_G_/I_D_*) can be regarded as a measure of the crystalline order/disorder of CNTs [33]. Among the tested samples, the raw CNT fiber showed the highest G to D ratio (*I_G_/I_D_* = 1.68). This value decreased with increasing reaction time (*I_G_/I_D_* = 1.48, 1.05, and 1.02 for 4 h, 8 h, and 12 h reactions, respectively), indicating a progressive degradation of the CNT crystalline structure in the CNT–MnO_2_ fibers. This was because carbon, acting as a reductant, was being consumed during the reaction with KMnO_4_ [34,35]. As a result, more defective CNTs were obtained with longer reaction times.

For CNT–MnO_2_ fibers, three major features originating from MnO_2_ can be recognized at ~500, 569, and 632 cm^−1^ (Figure 5b–d). The band at 632 cm^−1^ was due to the symmetrical Mn–O vibrations of the MnO_6_ groups, and the band located at 569 cm^−1^ can be attributed to the displacement of the oxygen atoms relative to the manganese atoms along the octahedral chains [28,36].

XPS measurements, undertaken to investigate the chemical composition of the fibers (Figure 6), revealed that the carbon-to-oxygen ratio (C/O) significantly decreased from 10.1 for the raw CNT fiber to 0.19 for the CNT–MnO_2_ composite fiber, implying the formation of metal oxides on the CNT surface. Figure 6b shows the deconvoluted high-resolution C 1 s spectrum from CNT–MnO_2_ fibers. The position and concentration of the fitted peaks are summarized in Table 1. It can be seen that the signal of the π–π^*^ shake-up feature totally disappeared in the CNT–MnO_2_ fiber. This observation indicates the formation of a thick MnO_2_ layer on the top of the CNT fiber [37]. The ratio of sp^2^ and sp^3^ concentrations was 4.9%, approximately two times smaller than that of the raw CNT fiber (9.3%). This decrease was in agreement with the observed loss of mechanical strength (Figure 4) and the observed decrease of the G to D ratio of the Raman peak intensities (Figure 5b–d). This observation also explains the reduced electronic conductivity of the composite fiber [38].

The oxidation state of the produced manganese oxide was characterized by analyzing the Mn 2p and Mn 3 s peaks in detail. The binding-energy separation between the two Mn 2p doublet peaks was 11.9 eV (Figure 6c), which closely matches the reported energy separation of MnO_2_ [39]. The analysis of the Mn 3 s peak led to the same conclusion: the doublet Mn 3 s peaks were caused by parallel spin coupling between electrons in the 3 s and 3d orbitals [25]. It has been proven that the 3 s peak energy separation is related to the oxidation states of Mn [18]. As can be seen from Figure 6d, the binding-energy separation between the two Mn 3 s doublet peaks was 4.7 eV, further suggesting that the manganese oxides were in the form of MnO_2_.

For the assembled supercapacitor, CNT–MnO_2_ fiber (10, 8) was used for the electrodes. As a good adhesion between the electrodes and the polymer electrolyte is of paramount importance for the supercapacitor performance, the wettability of the produced CNT–MnO_2_ fiber (10, 8) was tested with the tensiometric method. Details of the latter can be found in our previous work [29]. Interestingly, in this investigation, it was found that the CNT–MnO_2_ fiber was hydrophilic. By using the Cassie–Baxter model, equilibrium contact angles of a solid CNT–MnO_2_ fiber with the three test liquids (deionized water (DW), ethylene glycol (EG), and diidomethane (DIO)) were predicted, and the results are listed in Table 2. We previously reported a static water contact angle of 91° for CNT fiber [29], and Dubal et al. [40] reported that birnessite MnO_2_ film is hydrophilic with a static water contact angle of 21°. Thus, the hydrophilicity of the CNT–MnO_2_ fiber should be attributed to the decorating MnO_2_ nanoflakes rather than to CNTs. This hydrophilic property is strongly favorable for aqueous electrolyte (e.g., Na_2_SO_4_) ion transportation, allowing the electrolyte to access the active materials inside the fibers.

The surface energy components of CNT–MnO_2_ were calculated using the acid–base theory [41]. As can be seen in Table 3, the total surface tension of CNT–MnO_2_ (42.49 ± 2.32 mJ/m^2^) was larger than that of CNT (36.81 ± 2.59 mJ/m^2^), as was the basic energy component. This difference could have been caused by the addition of MnO_2_. Wetting parameters for CNT–MnO_2_ and common matrices for polymer electrolytes (PVA and PVDF) are listed in Table 4 (the surface energy components of PVDF were reported in our previous study [42] and the values for PVA were found in the literature, without standard deviation mentioned [43]). For both polymers, the spreading coefficient *S* significantly increases compared to that for CNT, which implies that spontaneous wetting of CNT–MnO_2_ is favored. PVA is more promising for achieving a better adhesion to CNT–MnO_2_, owing to its high *W_a_* and Δ*F* values. This is due to the fact that PVA has a similar *γ^LW^* to that of CNT–MnO_2_. As the adhesion between the electrodes and the polymer electrolyte plays a crucial role in the performance of the supercapacitor, we chose PVA as the matrix polymer electrolyte.

To test the electrochemical performance, a three-electrode cell configuration was employed by using CNT or CNT–MnO_2_ fibers for the working electrode (Figure 7). Details can be found in the experimental section. Electrochemical properties of the CNT–MnO_2_ fiber (10, 8) electrode are shown in Figure 8. The CV curve of the CNT–MnO_2_ fiber electrode at a scan rate of 30 mV/s is compared with that of the raw CNT fiber electrode (Figure 8a). The quasi-rectangular shape of the CV curves indicates good capacitive performance for both electrodes [44]. Nevertheless, the current density increased significantly for the CNT–MnO_2_ fiber electrode, owing to the pseudocapacitive contribution of MnO_2_. Figure 8b presents the CP curves of the CNT–MnO_2_ fiber (10, 8) electrode at various current densities. The charge/discharge time duration increased with the decreasing current density from 3 A/g to 1 A/g. Furthermore, the potential drop (IR drop), which is caused by the internal resistance of the electrode [45], became negligible by lowering the current density to 1 A/g.

The specific capacitance of the CNT–MnO_2_ fiber (10, 8) electrode is plotted versus various scan rates in Figure 8c. It shows that the capacitance decreased from 152 F/g to 92 F/g when the scan rate increased from 2 mV/s to 100 mV/s. Considering that the mechanism of the charge storage in MnO_2_-based electrodes can be described as [46]:
MnO_2_ + xA^+^ + xe^−^ ↔ A_x_MnO_2_(R1)
where *A* represents an alkali metal cation, the capacity drop can be explained as follows: The charge storage process is mainly governed by the insertion/extraction of Na^+^ and/or H^+^-ions from the electrolyte into/from the MnO_2_ nanoflakes. At faster scan rates, the Na^+^/H^+^ ions only reach the outer surface of the electrode [47]. As a result, the available capacity decreases with the increasing scan rate. In contrast, the specific capacitance of the CNT fiber electrode was determined to be only ~10 F/g, which is consistent with the values reported in the literature [16,48]. Moreover, the capacitance of the CNT fiber was unaffected by the scan rate, owing to the rapid charging/discharging of the electrochemical double layers.

In order to estimate the specific capacitance based on MnO_2_, the capacitance contribution of the CNTs needs to be subtracted from the total capacitance of the CNT–MnO_2_ fiber (Figure 8c). By taking into account that the total mass of the CNT–MnO_2_ fiber electrode originates from both, the CNTs and the deposited MnO_2_, the specific capacitance of MnO_2_ alone at 2 mV/s can be estimated as 265 F/g. Furthermore, at a scan rate of 100 mV/s, the specific capacitance of MnO_2_ is still as high as 158 F/g, indicating a high rate capacity of the electrode. So far, several studies [21,48,49,50,51] have reported supercapacitive performance in CNT–MnO_2_ electrodes [22]. Li et al. [49] reported that a CNT/MnO_2_ film electrode with 20 wt.% MnO_2_ delivered a specific capacitance of 151 F/g at a scan rate of 2 mV/s; Xie et al. [51] reported a specific capacitance of 205 F/g at a scan rate of 2 mV/s with a CNT–MnO_2_ pellet electrode, whereas only 43.2 F/g was measured at 50 mV/s. Compared with these previous studies, the CNT–MnO_2_ fiber electrode in our study exhibits a similar specific capacitance, but it shows an improved rate capability as well as having the advantage of flexibility. The good capacitive behavior in our wire-shaped electrode can be explained as follows: (i) the direct deposition of MnO_2_ nanoflakes on the CNT surfaces results in CNT–MnO_2_ fibers where the highly conductive CNT scaffold provides high electronic conductivity; (ii) the small size of the MnO_2_ nanoflakes promotes the pseudocapacitive reaction on the MnO_2_ surface, which ensures that a large fraction of the nanoflakes perform as active sites and guarantees a high specific capacitance and a good rate capability; (iii) the hydrophilic channels inside the CNT fiber facilitate penetration of the electrolyte into the inner spaces of the fiber and enhance the ionic conductivity of the electrode material. This CNT–MnO_2_ porous architecture enables the fiber electrode to have a fast electron and ion transport network, thus leading to excellent capacitive performance.

To estimate the specific capacitance of all the electrodes produced in this study, charge/discharge tests were performed between −0.1 V and 0.7 V. The results, summarized in Table 5, show that the specific capacitance of the electrodes increased with increased loading of MnO_2_, suggesting that it is mainly the pseudocapacitance of MnO_2_ that contributes to the capacitance. In addition, the measured values of MnO_2_-specific capacitance for the different CNT–MnO_2_ fiber electrodes were highly comparable, which justifies the method used to acquire the specific capacitance of MnO_2_. Based on these data, the CNT–MnO_2_ fiber (10, 8) was chosen for further study due to its relatively high specific capacitance and mechanical integrity.

Figure 8d shows Nyquist plots obtained from the raw CNT fiber and the CNT–MnO_2_ fiber (10, 8) electrodes analyzed by EIS. The Nyquist plots consist of (1) a high-frequency intercept on the real Z’ axis, (2) a semicircle arc in the high-to-medium-frequency, and (3) a low-frequency straight line [52]. The high-frequency intercept corresponds to the series resistance (*R_s_*), which mainly concerns the contribution of the electrolyte resistance, the intrinsic resistance of the active electrode material, and the contact resistance at the interface of the active material/current collector [53]. As can be seen in the insert of Figure 8d, the *R_s_* values for the CNT fiber and for the CNT–MnO_2_ fiber (10, 8) were ~35 Ω and 5 Ω, respectively. The large *R_s_* was probably caused by the contact resistance between the electrode and the current collector, as the fiber was too fine to have intimate contact with the current collector. The diameter of the semicircle corresponds to the charge transfer resistance (*R_ct_*) at the electrode/electrolyte interface [54]. The larger *R_ct_* of the CNT–MnO_2_ compared to the CNT fiber was due to the redox reaction of MnO_2_. The slope of the low-frequency line is an indication of a diffusion process (if the slope is 45°) or a capacitive process (if the slope is 90°) [55]. The nearly vertical line for the CNT fiber suggests good capacitive behavior, whereas the smaller slope for the CNT–MnO_2_ fiber indicates the cation diffusion resistance in MnO_2_, which follows from the charge storage mechanism (R1).

The specific capacitance of CNT–MnO_2_ fibers, derived from the voltametric response at a scan rate of 30 mV/s (Figure 8e), can be deconvoluted into a surface-limited process (∝ν, shaded area) and a diffusion-limited process, as described in the experimental section. From this data, we determined the total stored charge and the relative contributions associated with both diffusion-limited and surface-limited processes, as presented in Figure 8f. The surface-limited process led to a specific capacitance of about 90 F/g and was independent of the scan rate. The contribution from the diffusion-limited process, in contrast, strongly depended on the scan rate, decreasing from 33 F/g to 12 F/g when the scan rate increased from 10 mV/s to 50 mV/s. This is due to the fact that, at higher scan rates, the ionic insertion is limited and contributes less to the capacitance.

The fractional contribution of the surface-limited process was about 65% for CNT–MnO_2_ fiber (10, 8) at a scan rate of 4 mV/s, whereas it was 94% for CNT–MnO_2_ fiber (10, 4). This large difference may be attributed to thinner MnO_2_ flakes present in the CNT–MnO_2_ fiber (10, 4) and/or less dense loading of MnO_2_ flakes, leading to a larger specific surface area due to the larger area of exposed CNT surfaces.

In order to access the structural changes of the CNT–MnO_2_ fibers after electrochemical measurements, Raman spectroscopy was carried out (see Appendix A). The I_G_/I_D_ remained constant at 1.05 after testing, indicating the structural integrity of the CNT–MnO_2_ fiber, which is an important aspect for the cyclic stability of the electrode [56]. On the other hand, a band at ~650 cm^−1^ was present after testing. This peak is assigned to the Mn–O breathing vibration of Mn^2+^ in tetrahedral coordination, which indicates the existence of a Mn_3_O_4_ phase [36]. The formation of Mn_3_O_4_ could have been promoted by the immersion of the electrode in Na_2_SO_4_ solution [57].

Finally, a symmetric supercapacitor was assembled using two CNT–MnO_2_ (10, 8) fibers as electrodes and PVA/H_3_PO_4_ polymer as the solid electrolyte. The digital photos of the fabricated transparent and flexible supercapacitor are shown in Figure 9a. The supercapacitor had a length of ~5.5 cm and showed no breakage during bending. The ionic conductivity of the PVA/H_3_PO_4_ polymer electrolyte relies on the segmental motion of the PVA chain in the amorphous polymer phase [58]. X-ray diffraction (XRD) was used to examine the crystal structure of the synthesized polymer electrolyte (Appendix A). The pure PVA showed two peaks at the 2-theta angles of ~121° and 40°, corresponding to the semi-crystalline structure of PVA [59]. In contrast, the PVA/H_3_PO_4_ displayed much weaker and broader peaks, revealing the amorphous nature of the PVA/H_3_PO_4_ polymer electrolyte. Hence, the PVA/H_3_PO_4_ mass ratio of 1:2 used in this study can be considered as referring to a polymer electrolyte with higher ionic conductivity compared to pure PVA.

CV curves at different scan rates and CP curves at various current densities of the symmetric supercapacitor at a potential window of 0 to 1 V are displayed in Figure 9b,c. The nearly rectangular shape of the CV curves and the symmetric triangular shape of the CP curves were well maintained through the measurements, indicating the good capacitive behavior of the electrodes. However, a notable IR drop (0.1 V) was observed at the current density of 1 A/g, which can be ascribed to a non-negligible internal resistance of the device. The specific capacitance for the supercapacitor was 17.1 F/g at a current density of 1 A/g. This value was smaller compared to the capacitance determined by the three-electrode measurement (145 F/g), which was most probably caused by the high resistance of the polymer electrolyte. The supercapacitor showed a good cycling stability of almost 100% retention after 1200 cycles at 1 A/g (Figure 9d).

In general, the length-specific capacitance is utilized to evaluate the charge–discharge capacity of fiber-shaped supercapacitors [60]. As can be seen in Table 6, the length-specific capacitance decreased from 750 µF/cm to 622 µF/cm as the current density increased from 0.2 A/g to 1 A/g. This behavior was due to the insufficient time for the ions to approach the electrodes during fast charge/discharge processes. Notably, the length-specific capacitances of the supercapacitor are comparable with or even exceed those of many previously reported wire-shaped supercapacitors [1,2,21,24,61,62].

Energy and power density are two important parameters that help in evaluating the electrochemical performance of supercapacitors. Based on the CP curves, the energy density and power density were calculated, and they are displayed in a Ragone plot in Figure 10. The supercapacitor demonstrated a length energy density of 86 nWh/cm, while the length power density was 13 µW/cm at a current density of 1 A/g. The length energy density of our device surpassed the supercapacitors reported by Xu et al. [24] and Ren el al. [1], which are based on CNT–MnO_2_ fibers and CNT-OMC fiber, respectively. This high energy density can be attributed to the high and efficient loading of MnO_2_ achieved by immersing CNT fibers in KMnO_4_. However, the length specific power density was lower than the value reported in [24], which could be due to the high series and diffusive resistance in the device.

To investigate the influence of the electrolyte, two-electrode measurement was also performed with 1 M Na_2_SO_4_ bulk solution as the electrolyte. The supercapacitor delivered a specific capacitance of 44 F/g at 1 A/g, which corresponds to 176 F/g for a single electrode. This value is comparable with the results obtained from the three-electrode measurement (Table 5). Mechanical performance of the tested wire-shaped capacitor was examined (Appendix A) and a tensile strength of 274 MPa and strain-to-failure of 4.2% were found, which are comparable to cotton or silk fibers [28]. These results demonstrate that CNT–MnO_2_ fibers are promising candidates for electrodes in flexible supercapacitors and suggest that future work on device optimization is needed to improve their performance, in particular by addressing the problem of high resistance mentioned above.

In conclusion, MnO_2_ nanoflakes were homogeneously deposited onto CNT fibers through the direct redox reaction between CNTs and KMnO_4_. By tuning the concentration of the KMnO_4_ solution and the reaction time, different loadings of MnO_2_ were achieved. This very simple approach led to a homogeneous loading of CNT fibers with MnO_2_ nanoflakes of 2–3 nm in diameter. After decoration, CNT fibers changed from hydrophobic to hydrophilic while maintaining their flexibility and structural integrity. This simple one-step conversion was essential for the efficient loading of MnO_2_ nanoflakes, as well as to achieve sufficient adhesion between the electrodes and the electrolyte in the supercapacitor. The surface energy components of the MnO_2_-decorated CNT fibers were estimated by contact angle measurements. The CNT–MnO_2_ fiber electrode presented high capacitance, good rate performance, and a long cycle life in Na_2_SO_4_ solution. The contribution of the surface-limited process was as high as 65% for CNT–MnO_2_ fiber (10, 8).

In addition, a symmetric wire-shaped supercapacitor composed of CNT–MnO_2_ fiber electrodes and PVA/H_3_PO_4_ gel electrolyte was fabricated. The wire-shaped supercapacitor demonstrated a high length-specific capacitance of 621.8 µF/cm, with absolute cycle stability over 1200 cycles. The simple and tunable fabrication method for MnO_2_–CNT composite fibers reported in this work shows promising performance for use in the production of flexible and wearable fiber electrodes for future energy devices. Our detailed study also highlights the importance of the surface and mechanical properties of the fibers for the preparation and the performance of the wire-shaped supercapacitor, which have rarely been covered in the literature.

## 3. Experimental Section

Non-densified CNT fibers were used as a scaffold to form the composite electrode in this study. Individual CNTs have an average diameter of 15 nm with about 10 walls in average. Details about CNTs and the dry-spun CNT fibers can be found in our previous work [29]. The estimated densities of CNTs [16] and of the non-densified CNT fibers are 1.61 and 0.13 g/cm^3^, respectively. Thus, the calculated porosity of the non-densified CNT fiber is about 92%. 

For the preparation of CNT–MnO_2_ fiber electrodes, MnO_2_ was spontaneously deposited onto the CNT fibers through a direct redox reaction between the CNTs and KMnO_4_, according to the following chemical reaction [25,63]:4KMnO_4_ + 3C + H_2_O ⇌ 4MnO_2_ + K_2_CO_3_ + 2KHCO_3_(R2)

The procedure was as follows: (1) the raw CNT fiber was immersed in aqueous solutions (aq) with different concentrations of KMnO_4_ at 80 °C. (2) It was taken out after different reaction times and rinsed with deionized water to remove any excess solution. (3) The sample was annealed at 200 °C for 2 h to form CNT–MnO_2_ fiber. The weights of the raw CNT fiber and the CNT–MnO_2_ fiber were measured with a microbalance (Gamry Instruments, Warminster, PA, USA)with a weighing precision of ±10 μg. The effective mass loading of MnO_2_ was extracted from the weight difference based on (R2). The produced fibers are denoted “CNT–MnO_2_ fiber (c, t)”, where c stands for the KMnO_4_ solution concentration used (1 mM or 10 mM), and t represents the reaction time (4 h, 8 h, or 12 h).

For the fabrication of solid state supercapacitors, 1 g polyvinyl alcohol (PVA, Mw = 125,000 g/mol) was added to a mixed solution of 10 g deionized water and 2 g H_3_PO_4_ (85 wt.%). The mixture was constantly stirred at 85 °C until a transparent gel was formed. Two CNT–MnO_2_ fibers were coated with the gel electrolyte and dried at room temperature in a fume hood for 30 min. Subsequently, both fibers were carefully placed next to each other. The electrical resistance between these fibers was monitored with a multimeter to avoid any electrical short circuit. Finally, a thin layer of PVA/H_3_PO_4_ gel was applied again on the outside of the fibers to complete the fabrication of the wire-shaped solid state supercapacitor. The fabrication procedure of the wire-shaped supercapacitor is schematically illustrated in Figure 1.

CNT fibers were characterized with a scanning electron microscope (SEM, FEI Nova NanoSEM 450, Eindhoven, The Netherlands) equipped with an energy-dispersive X-ray spectrometer (Octane, EDAX, Mahwah, NJ, USA) (EDS). A probe-corrected transmission electron microscope (ARM200F cold-FEG, JEOL, Tokyo, Japan) was used to visualize the CNT and MnO_2_ structures (operated at an acceleration voltage of 200 kV) in the CNT–MnO_2_ composite. XPS studies were performed on a Thermo Fisher ESCALAB 250 Xi spectrometer (Waltham, MA, USA) equipped with an Al Kα monochromated X-ray source (1486.6 eV, spot diameter 500 μm). The charge neutralization of the sample was realized with a flood gun (Thermo Fisher, Waltham, MA, USA) using low-energy Ar^+^ ions and electrons. Samples were analyzed with a pass energy of 150 eV for survey spectra and 20 eV for high resolution scans. All binding energies were referenced to the adventitious C 1 s peak at 284.8 eV to correct the shift caused by charge effect. High resolution spectra were decomposed with a Gaussian/Lorentzian product function using XPS Peak 4.1 (freeware) (Shatin, Hong Kong, 2000). Raman spectroscopy was performed using an Ar-ion laser with the wavelength of 532 nm (Senterra, Bruker Optics, Billerica, MA, USA). The wettability of CNT–MnO_2_ fiber was studied with the tensiometric method [29] (Krüss K100SF) and surface energy components were derived by using the acid–base model [41,64]. The crystal structure of the PVA/H_3_PO_4_ solid polymer electrolyte was characterized by a Philips X’Pert X-ray diffractometer (XRD) (Eindhoven, The Netherlands) operating at 40 mA and 40 kV using monochromated Cu Kα1 radiation (wavelength = 1.54056 Å, step size = 0.2°).

Single-fiber tensile tests were performed according to the ASTM D3379 standard with a dynamic mechanical analyzer (Q800, TA Instruments, New Castle, DE, USA). A single CNT or CNT–MnO_2_ fiber was attached to a paper tab using adhesive. The paper tab was fixed with a tensile clamp and cut into two parts before testing (see Appendix A). A preload of 3 mN was applied at the beginning to stretch the fibers. The extension rate was set to 500 µm/min. An 18 N load cell with a resolution of 1 µN and a crosshead with a displacement resolution of 1 nm were used to record force and displacement. The gauge length was measured automatically before each test. At least five samples were tested for each type of fiber. 

In order to determine the tensile strength of CNT fibers, accurate measurement of the fiber cross-section areas is required. Cross-sectional area was calculated from the fiber diameter (assuming the fiber was a cylinder). Diameters measured at five different locations of the fiber by means of SEM were averaged. The force and displacement signals were then converted into stress and strain. The tensile strength of the fiber was taken as the maximum stress, and the stiffness was derived from the tangent slope of the stress–strain curve between 0.1% and 0.3% strain [33].

The electrochemical behaviors of the CNT fibers and CNT–MnO_2_ fibers were evaluated by cyclic voltammetry (CV), chronopotentiometry (CP), and electrochemical impedance spectroscopy (EIS) techniques on an Autolab electrochemical workstation at room temperature. A three-electrode cell configuration was employed, consisting of the raw CNT or CNT–MnO_2_ fiber as the working electrode, a Pt gauze as the counter electrode, a saturated calomel electrode (SCE) as the reference electrode, and 1 M Na_2_SO_4_ solution as the electrolyte (Figure 7). The fiber was clamped between current collectors (two pieces of graphite). CV measurements were carried out between −0.1 V and 0.7 V (vs. SCE) at various scan rates ranging from 2 mV/s to 100 mV/s (in this voltage window the CV curves were rectangular, while an unsuitable voltage window led to a deformed CV curve, see, e.g., Appendix A). The CP was also conducted in the same voltage window at different specific currents ranging from 1 A/g to 3 A/g. EIS measurements were carried out in a frequency range from 10 kHz to 0.01 Hz at open circuit potential with an amplitude of 5 mV. 

The electrochemical performances of the supercapacitor were tested in a two-electrode configuration in the voltage range of 0 V and 1 V. In the three-electrode system, the average specific capacitance of the electrodes was calculated from the CV diagrams according to Equation (1) [63]:*C_s,electrode_* = (*q_a_* + |*q_c_*|)/2*m*∆*V*(1)
where *C_s,electrode_(F/g)* is the mass specific capacitance, *m(g)* is the mass of the active material on the electrode, and ∆*V(V)* is the voltage window. *q_a_* and *q_c_* are the anodic and cathodic charges in coulombs, respectively. The *C_s,electrode_* was calculated from the CP curves by using Equation (2):*C_s,electrode_* = *I* ∆*t*/(*m* ∆*V*)(2)
where *I(A)* is the discharge current and ∆*t(s)* is the discharge time. In the two-electrode system, the *C_s,electrode_* was derived from Equation (3)
*C_s,electrode_* = 4*I* ∆*t*/(*M* ∆*V*)(3)
where *M(g)* is the total mass of the active materials on the positive and negative electrodes, with *M* = 2*m* in a symmetric supercapacitor. The energy and power densities of supercapacitors were calculated from:*E_device_* = *C_device_ V*^2^/2(4)
*P_device_* = *E_device_*/2 ∆*t*.(5)
where *E_device_*(Wh/kg) is the energy density, *C_device_*(F/g) is the capacitance of the supercapacitor, *V*(V) is the cell voltage, and *P_device_*(W/kg) is the power density. The mass of the active material *m* in the above equations can be replaced with the length of the electrode to acquire the average length-specific capacitances *C_length_*.

Taking (R1) into account, the total stored charge can be separated into three components: (i) the Faradaic contribution from the alkali metal cations (*A*^+^) insertion process, (ii) the Faradaic contribution from the charge-transfer process with surface atoms, i.e., pseudocapacitance, and (iii) the non-Faradaic contribution from the double layer charging. The first component is a diffusion-limited process while the latter two components are surface-limited processes. These capacitive effects can be characterized by analyzing the CV data at various scan rates according to [65]:*i = aυ^b^*(6)
where *i* is the measured current, *a* is a material-dependent constant, *υ* is the scan rate, and *b* is a parameter that relates to the charge storage mechanism. For voltametric processes controlled by diffusion, *b* = 0.5; for non-insertion capacitive processes, *b* = 1. Consequently, the measured current *i* at a fixed potential *(V)* can be expressed as [66]:*i (V)* = *k*_1_*υ* + *k*_2_*υ*^1/2^(7)
where *k*_1_ and *k*_2_ are scan rate-independent constants. *k*_1_*υ* and *k*_2_*υ*^1/2^ correspond to the current contributions from the surface-limited process and the diffusion-limited process, respectively. Equation (7) can be transformed into:*i (V)*/*υ*^1/2^ = *k*_1_*υ*^1/2^ + *k*_2_(8)

By plotting the scan rate dependence of the current according to Equation (8), the values of *k*_1_ and *k*_2_ were obtained from the slope and the *y*-axis intercept, respectively.

## Figures and Tables

**Figure 1 molecules-26-03479-f001:**
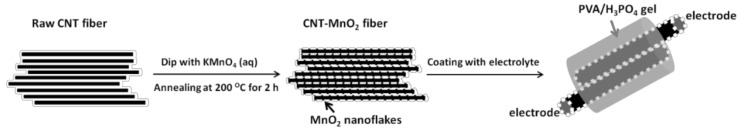
Schematic diagram of the manufacturing method of the CNT–MnO_2_ supercapacitors.

**Figure 2 molecules-26-03479-f002:**
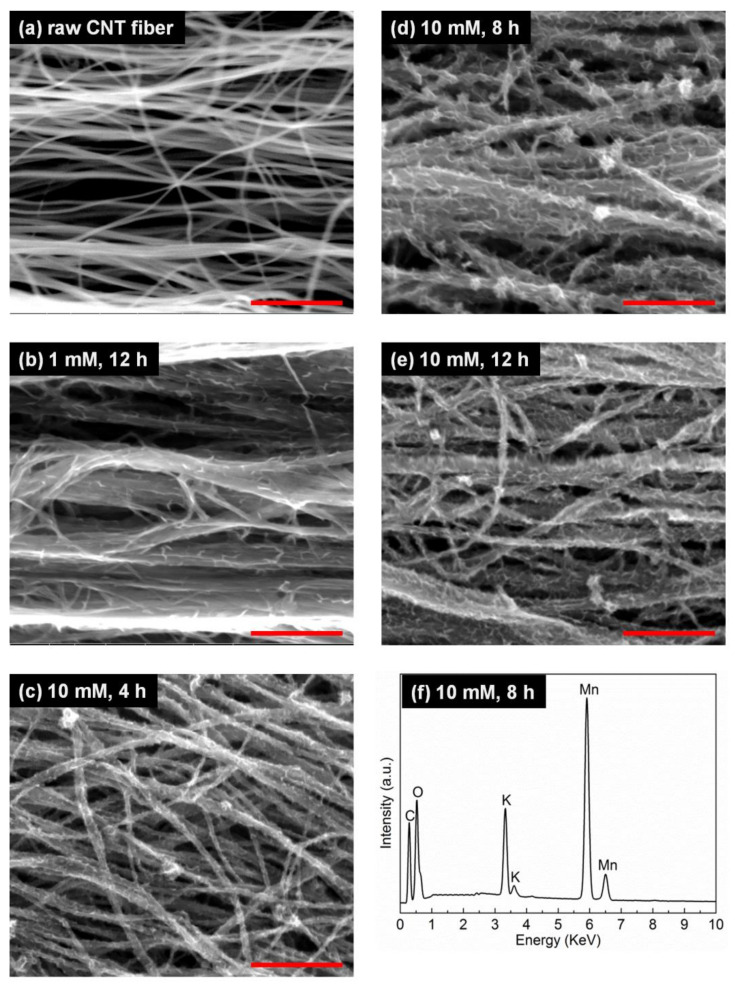
SEM images of (**a**) raw CNT fiber and (**b**) CNT–MnO_2_ fiber (1, 12), (**c**) (10, 4), (**d**) (10, 8), and (**e**) (10, 12). (**f**) EDS pattern detected from CNT–MnO_2_ fiber (10, 8). Scale bars equal 500 nm.

**Figure 3 molecules-26-03479-f003:**
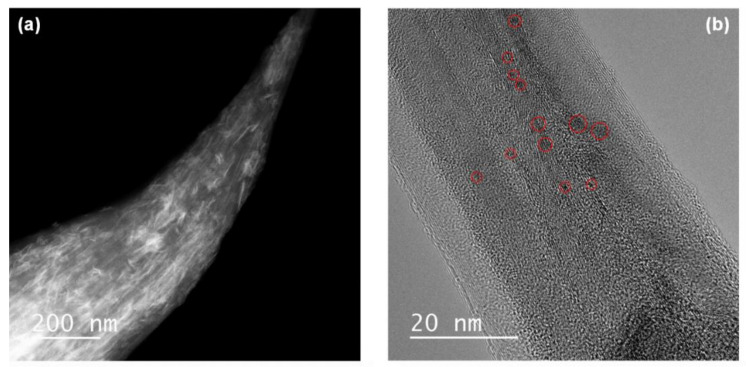
(**a**) ADF scanning TEM image of CNT–MnO_2_ fiber (10, 8) showing that MnO_2_ nanoflakes (bright contrast) were present along the CNT fiber. (**b**) High-resolution TEM image revealing that the size of the MnO_2_ nanoflakes (some are marked by red circles) was approximately 2–3 nm.

**Figure 4 molecules-26-03479-f004:**
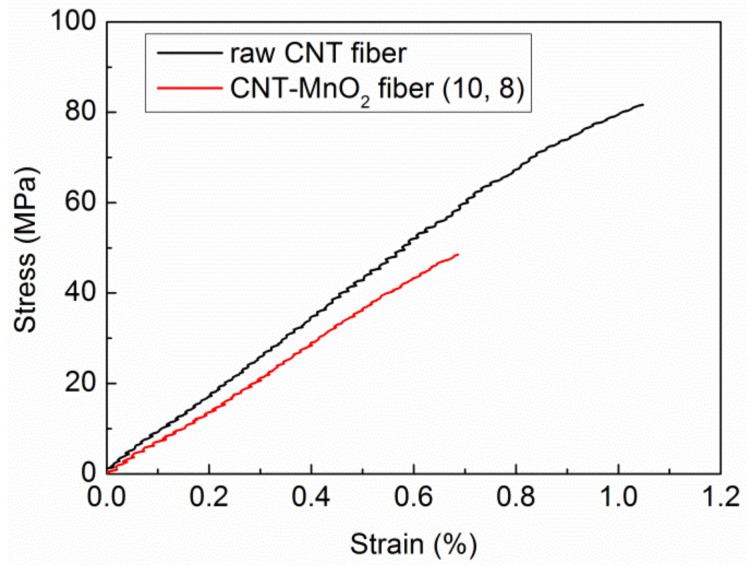
Strength–strain curves of single fiber tensile test.

**Figure 5 molecules-26-03479-f005:**
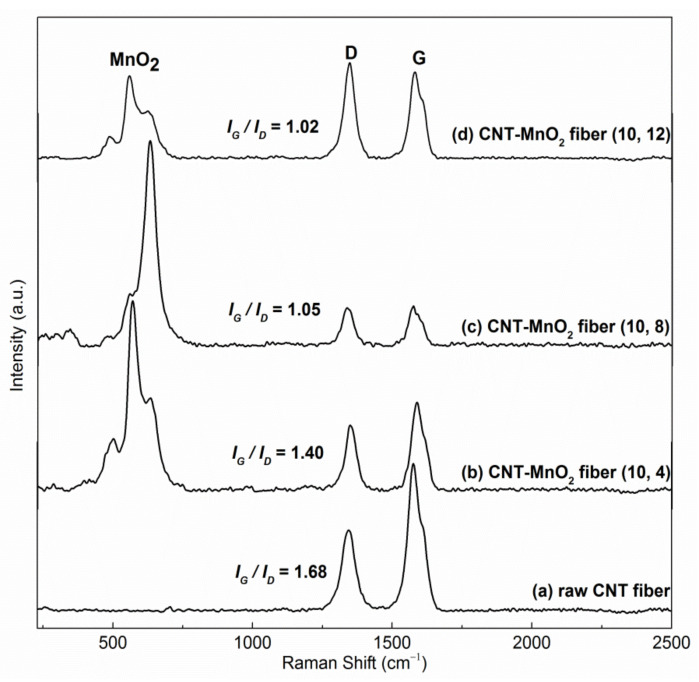
Raman spectra of (**a**) raw CNT fiber and (**b**) CNT–MnO_2_ fibers (10, 4), (**c**) (10, 8), and (**d**) (10, 12).

**Figure 6 molecules-26-03479-f006:**
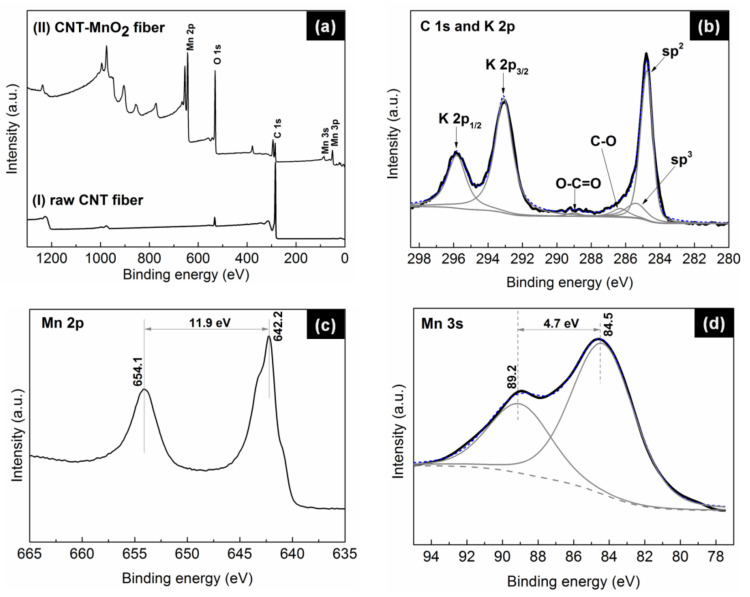
XPS survey spectra of (**a**) CNT fiber (bottom) and CNT–MnO_2_ fiber (top). High-resolution XPS spectra from CNT–MnO_2_ fiber: (**b**) C 1 s and K 2p, (**c**) Mn 2p, and (**d**) Mn 3 s peaks. The splitting width of the doublet peaks in Mn 3 s was 4.7 eV.

**Figure 7 molecules-26-03479-f007:**
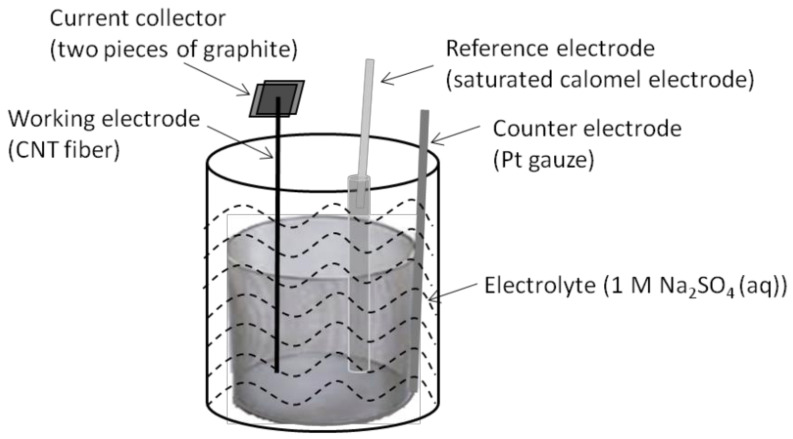
Schematic representation of the three-electrode cell used to test the electrochemical performance of individual CNT–MnO_2_ fibers.

**Figure 8 molecules-26-03479-f008:**
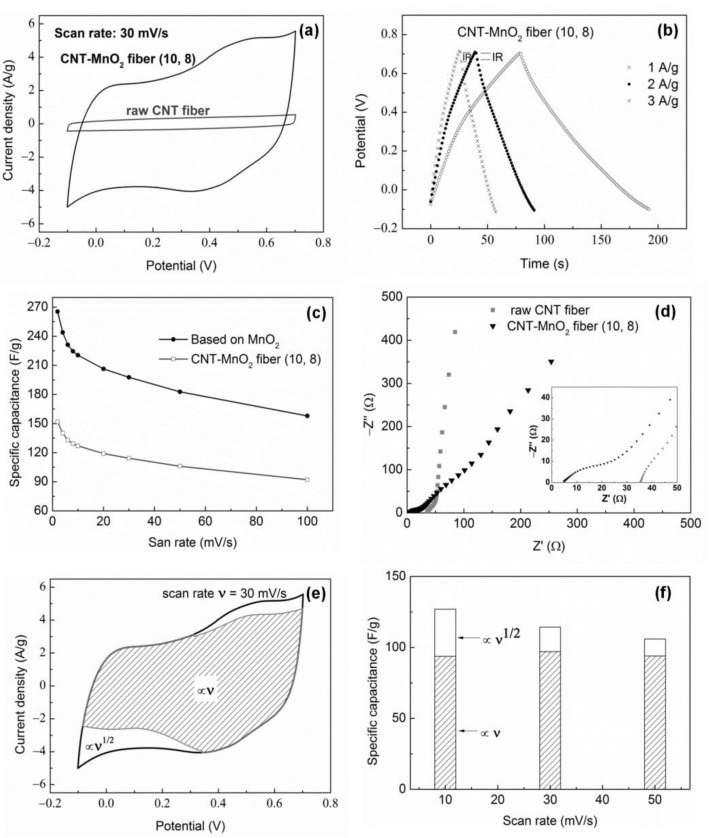
Electrochemical characterization of CNT–MnO_2_ fiber electrode (working electrode) in a three-electrode cell using 1M Na_2_SO_4_ (aq) as electrolyte: (**a**) comparison of CV curves of raw CNT fiber and CNT–MnO_2_ fiber (10, 8) electrodes; (**b**) CP curves of CNT–MnO_2_ fiber (10, 8) electrode at various current densities; (**c**) specific capacitance for the CNT–MnO_2_ fiber (10, 8) electrode and the deposited MnO_2_ at different scan rates; (**d**) Nyquist plots of CNT fiber and CNT–MnO_2_ fiber (10, 8) electrodes. (**e**,**f**) Deconvolutions of two contributions to the capacitance in CNT–MnO_2_ fiber (10, 8): the diffusion-limited process (∝ν^1/2^, blank regions) and the surface-limited process (∝ν, shaded regions). (**e**) CV at 30 mV/s and (**f**) bar graph of the two contributions versus the scan rate ν.

**Figure 9 molecules-26-03479-f009:**
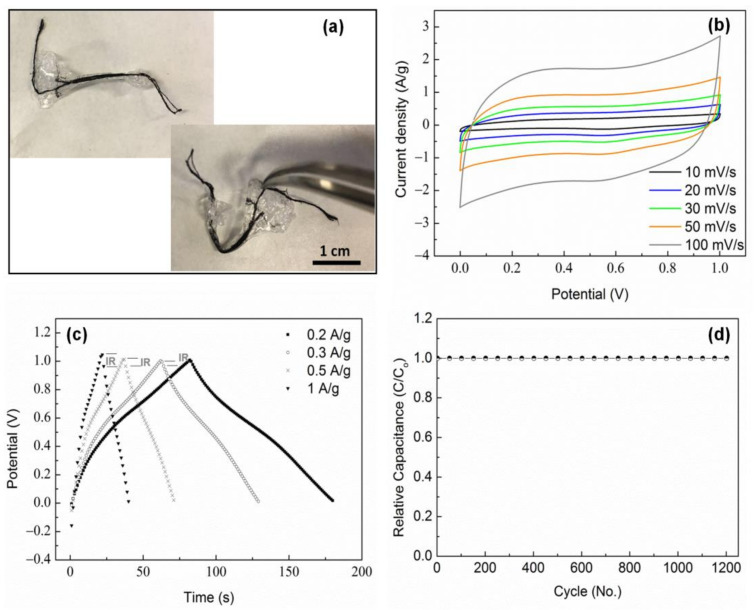
Electrochemical characterization of the symmetric CNT–MnO_2_ fiber supercapacitor using PVA/H_3_PO_4_ polymer as the solid electrolyte: (**a**) photograph showing the flexibility of the wire-shaped supercapacitor, (**b**) CV curves of the CNT–MnO_2_ fiber supercapacitor at various scan rates, (**c**) CP curves at different current densities, and (**d**) cycling performance of the supercapacitor at a current density of 1 A/g.

**Figure 10 molecules-26-03479-f010:**
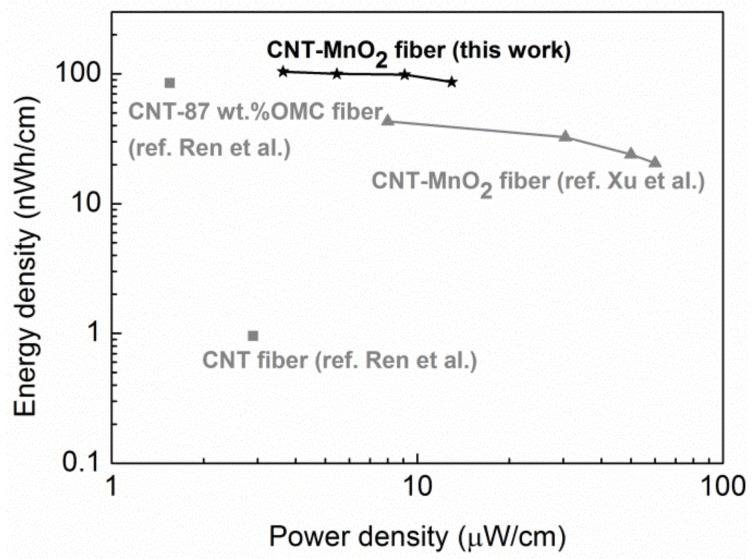
Ragone plots of length energy density versus length power density for the CNT–MnO_2_ fiber supercapacitor fabricated in this work (black star); asymmetric supercapacitors based on pristine and MnO_2_-coated CNT fibers (grey triangle) [24]; and supercapacitors based on bare CNT fiber and CNT-OMC fibers (grey square) [1].

**Table 1 molecules-26-03479-t001:** Surface functional components obtained from the deconvolution of the C 1 s peak for CNT–MnO_2_ fiber.

Assignment	sp^2^ Carbon	sp^3^ Carbon	C-O	O-C=O	K 2p_3/2_	K 2p_1/2_
Binding energy (eV)	284.8	285.4	286.3	289.0	293.1	295.9
Concentration (%)	30.4	6.2	3.0	1.1	40.4	19.0

**Table 2 molecules-26-03479-t002:** Advancing, receding, and equilibrium contact angles measured for DW, EG, and DIO. Equilibrium contact angles of the solid CNT–MnO_2_ fiber (10, 8) (*θ*’_*equ(CNT-MO2)*_) were calculated by using the modified Cassie–Baxter model.

	CNT–MnO_2_ Fiber				CNT–MnO_2_
Test Liquid	*θ_adv_* (°)	*θ_rec_* (°)	*θ_equ_* (°)	*θ_hyst_* (°)	*θ*’*_equ(CNT-MO2)_* (°)
DW	87.3 ± 3.3	32.2 ± 5.0	63.4 ± 4.2	55.1	69.9 ± 3.8
EG	39.2 ± 5.6	34.4 ± 6.1	36.9 ± 5.8	4.9	46.2 ± 5.3
DIO	39.5 ± 5.7	19.3 ± 2.3	24.9 ± 4.4	10.2	34.9 ± 4.3

**Table 3 molecules-26-03479-t003:** Surface energy components of CNT and CNT–MnO_2_ calculated according to the acid–base model.

Material	*γ* (mJ/m^2^)	*γ^LW^* (mJ/m^2^)	*γ^ab^* (mJ/m^2^)	*γ*^+^ (mJ/m^2^)	*γ*^−^ (mJ/m^2^)
CNT	36.81 ± 2.59	35.88 ± 2.51	0.94 ± 0.63	0.16 ± 0.18	1.36 ± 1.02
CNT–MnO_2_	42.49 ± 2.32	42.07 ± 1.99	0.41 ± 1.19	0.01 ± 0.05	4.87 ± 1.64

**Table 4 molecules-26-03479-t004:** Wetting parameters *W_a_*, Δ*F*, *S*, and *γ*_sl_ predicted based on the surface energy components of CNT and CNT–MnO_2_.

Matrix	Substrate	*W_a_* (mJ/m^2^)	*S* (mJ/m^2^)	*γ*_sl_ (mJ/m^2^)	Δ*F* (mJ/m^2^)
PVDF	CNT	70.56 ± 2.20	1.28 ± 1.14	0.89 ± 0.92	35.92 ± 1.67
	CNT–MnO_2_	77.01 ± 2.47	7.73 ± 1.41	0.12 ± 0.38	42.37 ± 1.94
PVA	CNT	80.93	−3.07	−2.12	38.93
	CNT–MnO_2_	84.87	0.87	−0.38	42.87

**Table 5 molecules-26-03479-t005:** Capacitance values for raw CNT fiber and CNT–MnO_2_ fiber electrodes derived from CP detected in 1 M Na_2_SO_4_.

Material	KMnO_4_ (aq) Concentration (mM)	Reaction Time (h)	MnO_2_ Loading Percentage (%)	Specific Capacitance (F/g) at 1 A/g	Specific Capacitance Based on MnO_2_ (F/g) at 1 A/g
Raw CNT fiber		/	0	10.5	
CNT–MnO_2_ fiber	10	4	42.7	122.5	264.9
	10	8	55.7	145.0	252.4
	10	12	85.7	231.3	268.1

**Table 6 molecules-26-03479-t006:** Comparison of specific capacitances of wire-shaped supercapacitors.

Electrode Material	Length-Specific Capacitance (µF/cm)	Characterization Conditions
This study	621.8–749.1	CP: 0.2–1 A/g
CNT–MnO_2_ fiber [21]	16–19	CP: 0.5–10 µA
CNT–MnO_2_ fiber and CNT fiber [24]	113.21–157.29	CP: 0.6–5.7 mA/cm^2^
TiO_2_ nanotubes on wire and CNT sheet [61]	156	CP: 1 µA
Reduced graphene oxide on Au wire [62]	11.4	CP: 2.5 µA/cm
CNT-ordered mesoporous carbon (OMC) fiber [1]	750–953.5	CP: 0.5–10 µA

## Data Availability

The data presented in this study are available on request from the corresponding authors.

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
