# Peer review of "Carbon Nanotube Fibers Decorated with MnO2 for Wire-Shaped Supercapacitor"

_molecules, 2021, doi:10.3390/molecules26113479_

Round 1
Reviewer 1 Report
In this work, the authors clearly describe how the loading of CNT fibers using MnO2 nanoflakes can increase their electrochemical properties. The presented CNT-MnO2 electrodes have been tested extensively using optical, mechanical, and electrochemical methods. Various loading concentrations have been compared successfully. The manuscript is very well structured and written. However, some minor comments should be considered before accepting this article for publication.
- In the abstract and introduction, it is not quite clear how far improved electrochemical properties of CNTs could benefit the scientific or industrial community. It would be very helpful to give some examples of applications of CNTs (e.g. solar collection, neural stimulation, coatings, etc.). Thereby the importance of this work could be highlighted.
- When performing CV, CP, and EIS, the electrolyte was a 1 M Na2SO4 solution. Did the authors consider the application of the here presented CNT-MnO2 fiber as a physiological setup (i.e. phosphate buffer saline solution)? Discussing this point in the manuscript would add importance to the work and may promote their distribution.
- Page 16, line 502 to 506: there seems to be a format issue. The flow-text has a larger font size compared to the other.
- The font of figures: 2,5,6,8,9,10 containing plots is quite small compared to the flow-text and could be increased
Author Response
We sincerely thank the reviewer for the valuable comments. For your convenience, our comments are inserted in black whereas the comments of the reviewer appear in blue.
In this work, the authors clearly describe how the loading of CNT fibers using MnO2 nanoflakes can increase their electrochemical properties. The presented CNT-MnO2 electrodes have been tested extensively using optical, mechanical, and electrochemical methods. Various loading concentrations have been compared successfully. The manuscript is very well structured and written. However, some minor comments should be considered before accepting this article for publication.
- In the abstract and introduction, it is not quite clear how far improved electrochemical properties of CNTs could benefit the scientific or industrial community. It would be very helpful to give some examples of applications of CNTs (e.g. solar collection, neural stimulation, coatings, etc.). Thereby the importance of this work could be highlighted.
Following the suggestion of the reviewer, we have added two sentences in the abstract as follows: “Fibers made from CNTs (CNT fibers) have the potential to form high-strength, lightweight materials with superior electrical conductivity. CNT fibers have attracted great attention for various applications, in particular as conductive electrodes in energy applications, such as capacitors, lithium-ion batteries and solar cells.”
In addition, two sentences have been added in the introduction to better highlight the potential of CNT fibers for electrochemical devices, including five new references: “The outstanding potential of CNT fibers for various fiber-based devices have been demonstrated, including e.g. supercapacitors [10], solar cells [11], actuators [12] and biosensors [13]. CNT fiber electrodes have also been applied for biocompatible implantable supercapacitors [14] as well as for neurochemical monitoring [15].”
- When performing CV, CP, and EIS, the electrolyte was a 1 M Na2SO4 solution. Did the authors consider the application of the here presented CNT-MnO2 fiber as a physiological setup (i.e. phosphate buffer saline solution)? Discussing this point in the manuscript would add importance to the work and may promote their distribution. Considering of the potential use in physiological setup, a neutral solution of 1 M Na2SO4 solution were used as electrolyte for performing CV, CP, and EIS test.
We thank the reviewer for this interesting suggestion. For the current manuscript, we have limited our work on the supercapacitor application and have not considered to use CNT-MnO2 fibers in a physiological setup. This could be explored in a future work. However, there are so far only very few literatures involving MnO2 in implantable electrodes. This might be due to the fact that Mn is toxic to various cell types. In that sense, CNT-MnO2 fibers could be not the best candidate for biological and/or biomedical applications.
- Page 16, line 502 to 506: there seems to be a format issue. The flow-text has a larger font size compared to the other.
The font issue has been solved.
- The font of figures: 2,5,6,8,9,10 containing plots is quite small compared to the flow-text and could be increased
For the respective figures, the font size has been increased.
Reviewer 2 Report
This work reports on the dry-spun CNT fibers from CNT carpets homogeneously loaded with MnO2 nanoflakes by treating with KMnO4. The MnO2-CNT fiber electrode was reported with a specific capacitance of 231.3 F/g in Na2SO4 electrolyte. The symmetric wire-shaped supercapacitor was demonstrated with an energy density of 86 nWh/cm and good cycling performance. The CNT-based wire-shaped supercapacitor shows promise for its application in flexible and wearable energy storage devices. Even though the manuscript is well organized, it needs to be revised with major corrections before publication in this reputed journal:
- It is recommended that the author carefully check the grammatical error in the manuscript and make necessary corrections.
- The novelty of the work should be highlighted in detail compared to existing similar reports.
- The performance of the MnO2-CNT fiber as the negative electrode should be provided.
- What is the reason to represent energy density and power density with respect to area and other terms, current density, specific capacitance, etc., with respect to mass? The representation of units should be reorganized uniformly throughout the manuscript.
- Why is the voltage range differs with respect to the applied current density in Fig. 8b and Fig. 9c.
- CV curves under different operating voltage windows should be given in order to determine the suitable operating windows.
- Some recent reference can be cited: 10.3390/polym10101152; 10.1016/j.apsusc.2020.145424; 10.1515/hf-2017-0143; 10.3390/en12163127; 10.1016/j.apsusc.2019.145157;
Author Response
We sincerely thank the reviewer for the valuable comments. For your convenience, our comments are inserted in black whereas the comments of the reviewer appear in blue.
This work reports on the dry-spun CNT fibers from CNT carpets homogeneously loaded with MnO2 nanoflakes by treating with KMnO4. The MnO2-CNT fiber electrode was reported with a specific capacitance of 231.3 F/g in Na2SO4 electrolyte. The symmetric wire-shaped supercapacitor was demonstrated with an energy density of 86 nWh/cm and good cycling performance. The CNT-based wire-shaped supercapacitor shows promise for its application in flexible and wearable energy storage devices. Even though the manuscript is well organized, it needs to be revised with major corrections before publication in this reputed journal:
- It is recommended that the author carefully check the grammatical error in the manuscript and make necessary corrections.
We carefully checked the manuscript for grammatical errors.
- The novelty of the work should be highlighted in detail compared to existing similar reports.
In order to position the novelty of this work better, we have added the following sentence into the second last paragraph of the introduction:
“This systematic study also demonstrates that various aspects have a significant impact on the surface and mechanical properties of CNT-MnO2 fibers, which have so far not been considered sufficiently in the literature and pinpoint that a particular attention is required when preparing the fibers and supercapacitors.”
Moreover, in conclusion (last two paragraphs of Results and Discussion, on page 14) we have added two sentences
- highlighting the simple and efficient procedure of CNT-MnO2 fiber preparation “This very simple approach leads to a homogeneous loading of CNT fibers with MnO2 nanoflakes of 2 - 3 nm in diameter. After decoration, CNT fibers changed from hydrophobic to hydrophilic while maintaining their flexibility and structural integrity. This simple one-step conversion is essential for the efficient loading of MnO2 nanoflakes as well as for the sufficient adhesion between the electrodes and the electrolyte in the supercapacitor.”
- pinpointing our systematic study including surface and mechanical properties of the CNT-MnO2 fibers “Our detailed study also highlights the importance of surface and mechanical properties of the fibers for the preparation and the performance of the wire-shaped supercapacitor, which have only rarely been covered in the literature.”
- The performance of the MnO2-CNT fiber as the negative electrode should be provided.
Please consider that in the two electrode setup (Figure 9), MnO2-CNT fibers were used as both negative and positive electrode.
- What is the reason to represent energy density and power density with respect to area and other terms, current density, specific capacitance, etc., with respect to mass? The representation of units should be reorganized uniformly throughout the manuscript.
We appreciate your comment. However, for fiber-based devices, the length and area are more commonly used for the evaluation of energy density and power density due to the practical consideration (see the references [1] and [5]). To compare the performance of the CNT-MnO2-fiber supercapacitor with that of other studies reporting CNT fiber supercapacitors (Table 6), we converted some of the values and plotted all results uniformly as length specific energy density and power density.
- Why is the voltage range differs with respect to the applied current density in Fig. 8b and Fig. 9c.
The reason is that in our measurement system setup, 4 data points were still recorded after exceeding the voltage window. The higher the current density, the larger the ‘over-(dis-)charge’. Figure 8b and 9c were revised by removing these extra data points.
- CV curves under different operating voltage windows should be given in order to determine the suitable operating windows.
Thank you for the valuable comment. We have added in the experimental section an explanation why the -0.1-0.7V (vs. SCE) voltage window was chosen for the three-electrode cell. The figure shown below is added as Fig. S5 into the supplementary information. The shape of CV curves deviates from a rectangle when using a broader voltage window. Therefore, the voltage range -0.1-0.7V (vs. SCE) is used for the three-electrode test.
Figure S5. A non-quasi-rectangular CV curve of a CNT-MnO2 fiber electrode, operating voltage window: -0.2 – 0.8V.
- Some recent reference can be cited: 10.3390/polym10101152; 10.1016/j.apsusc.2020.145424; 10.1515/hf-2017-0143; 10.3390/en12163127; 10.1016/j.apsusc.2019.145157;
We thank the reviewer for the valuable suggestions. However, we were not able to cite all the references mentioned above as they are not related to CNT fiber based capacitors. In order to include more recent, directly relevant works, we added the references [2], [15], [17] and [20] into the manuscript.
[2] Lu, Z.; Raad, R.; Safaei, F.; Xi, J.; Liu, Z.; Foroughi J. Carbon Nanotube Based Fiber Supercapacitor as Wearable Energy Storage. Front. Mater. 2019, 6, 138.
[15] Tan, C.; Robbins, E. M.; Wu, B.; Cui X. T. Recent Advances in In Vivo Neurochemical Monitoring. Micromachines 2021, 12, 208.
[17] Sim, Y.; Kim, S. J.; Janani, G.; Chae, Y., Surendran S.; Kim, H.; Yoo, S.; Seok, D. C.; Jung, Y. H.; Jeon, C.; Moon, J.; Sim, U. The synergistic effect of nitrogen and fluorine co-doping in graphene quantum dot catalysts for full water splitting and supercapacitor. 2020, 507, 145157.
[20] Wu, D.; Xie, X.; Zhang, Y.; Zhang, D.; Du W.; Zhang, X.; Wang, B. MnO2/Carbon Composites for Supercapacitor: Synthesis and Electrochemical Performance. Front. Mater. 2020, 7, 2.

Reviewer 3 Report
The manuscript reports interesting results concerning MnO2-modified CNT preparation and emphasizes the promising capacitive features of the CNT fibers thus functionalized.
The active material was thoroughly characterized, the electrochemical experiments were all conducted, the results were clearly reported and seem quite reliable. In my opinion, the work is worthy of publishing in this journal.
Author Response
We sincerely thank the reviewer for the positive feedback.
Round 2
Reviewer 2 Report
The revised manuscript has been greatly improved by carefully addressing the reviewers' suggestions. I kindly recommend its publication.